# A Novel Intraoperative Mapping Device Detects the Thermodynamic Response Function

**DOI:** 10.3390/brainsci13071091

**Published:** 2023-07-19

**Authors:** Michael Iorga, Nils Schneider, Jaden Cho, Matthew C. Tate, Todd B. Parrish

**Affiliations:** 1Department of Radiology, Feinberg School of Medicine, Chicago, IL 60611, USA; 2Department of Neurological Surgery, Feinberg School of Medicine, Chicago, IL 60611, USA; 3Department of Neurology, Feinberg School of Medicine, Chicago, IL 60611, USA; 4Department of Biomedical Engineering, Northwestern University, Evanston, IL 60208, USA

**Keywords:** infrared thermography, intraoperative mapping, functional neuroimaging, brain mapping, image-guided neurosurgery

## Abstract

Functional activation leads to an increase in local brain temperature via an increase in local perfusion. In the intraoperative setting, these cortical surface temperature fluctuations may be imaged using infrared thermography such that the activated brain areas are inferred. While it is known that temperature increases as a result of activation, a quantitative spatiotemporal description has yet to be achieved. A novel intraoperative infrared thermography device with data collection software was developed to isolate the thermal impulse response function. Device performance was validated using data from six patients undergoing awake craniotomy who participated in motor and sensory mapping tasks during infrared imaging following standard mapping with direct electrical stimulation. Shared spatiotemporal patterns of cortical temperature changes across patients were identified using group principal component analysis. Analysis of component time series revealed a thermal activation peak present across all patients with an onset delay of five seconds and a peak duration of ten seconds. Spatial loadings were converted to a functional map which showed strong correspondence to positive stimulation results for similar tasks. This component demonstrates the presence of a previously unknown impulse response function for functional mapping with infrared thermography.

## 1. Introduction

Resective surgery is a fundamental part of glioma management [1,2,3]. Increased extent of tumor resection has been shown to increase both patient survival and functional outcomes [4,5]. As diffuse tumors, gliomas may extend beyond the radiographic boundaries and infiltrate healthy tissues across the entire brain. It may be appropriate in some cases to resect additional tissues to limit the possibility of recurrence [6]. However, increasing the resection area also increases the risk of postoperative neurological deficits [7]. Even though the location of functional brain regions are approximated based on known canonical network nodes or presurgical mapping studies, there is significant variation in functional anatomy among patients, particularly those harboring infiltrating tumors which have been shown to induce functional neuroplasticity [8]. It is therefore imperative, and standard of care, that intraoperative functional mapping be conducted alongside resective surgery to identify and preserve these eloquent areas in cases where tumor resection is being performed in the vicinity of presumed functional brain regions.

Direct electrical stimulation (DES) is the current gold standard for functional mapping during glioma surgery [9]. There is strong evidence showing that intraoperative mapping with DES can prevent postoperative functional deficits without impeding extensive resection [10]. DES can be performed at both the cortical and subcortical levels [11]. DES-based mapping is flexible and has been used to map a variety of functions including motor, sensory, language, and cognition [12]. The primary disadvantages of DES are that it can only interrogate one area at a time and that it has a relatively low spatial resolution of 5 mm. Stimulation-induced seizures occur in about four percent of patients [13] and are a major contributor to mapping failure. The gross effects of stimulation on cortical tissue are complex and overall poorly understood [14]. Accurate DES current thresholds may differ significantly between different brain areas in a single subject [15]. Simulation may incidentally interact with local inhibitory circuits, leading to false-negative mapping results [16]. These limitations have raised interest in alternative mapping techniques.

Infrared thermography (IRT) is an experimental approach for intraoperative functional mapping which identifies eloquent areas through stimulus-dependent changes in brain surface temperature [17,18,19]. Activation of functional areas leads to an increase in local perfusion, which during craniotomy leads to heating on the brain surface [20,21]. Gorbach et. al. demonstrated agreement between IRT and DES across motor, sensory, and language tasks in awake patients and specifically identified a sharp temperature increase of 0.04–0.08 °C five to seven seconds following functional activation [22]. These temperature values were corroborated with experiments in a rodent model demonstrating temperature increases of 0.05–0.10 °C in the barrel cortex in response to whisker stimulation [23]. Further investigation in rodents with simultaneous thermal and laser speckle imaging demonstrated that these temperature increases are induced by local changes in cerebral blood flow [24]. The triangulation between functional activation, hemodynamics, and thermodynamics suggests that cortical temperature may operate as a functional contrast similar to the functional MRI BOLD signal.

As an optical method, IRT has several distinct advantages versus DES. Foremost, IRT is able to simultaneously map the entire exposed craniotomy, which may lead to faster mapping and interrogation of cortical networks in a way that is not possible with single localized DES stimulations. This also enables a different approach to awake task design. Whereas DES revolves around a nonphysiological elicit or interrupt paradigm, in IRT mapping, the subject instead simply performs the function of interest and the resulting physiological changes are observed. IRT leverages advanced thermal imaging technology to deliver spatial resolutions as low as 100 μm. Finally, IRT is noncontact and incurs no additional safety risks such as intraoperative seizures. However, there are also some weaknesses. First, IRT relies on temperature differences which are difficult to discern without a dedicated imaging system for data analysis and visualization. Second, infrared signal attenuation by water limits mapping to the brain surface, so deeper structures or any areas beyond the edge of the craniotomy cannot be observed. Third, the temperature spread of IRT is broader than the critical network nodes which may lower specificity as compared to DES. Finally, it is currently difficult to differentiate eloquent areas from other tissues with high confidence. Although the literature states that there are functionally induced temperature changes, the precise structure and timing of these changes remain unclear.

The primary goal of this work is to address the present limitations of IRT and modernize the technical framework for IRT-based mapping. We have developed a novel infrared thermography system for awake functional mapping with custom software and integrated devices for task administration and behavioral monitoring. We describe the major sources of artifacts in thermography data and preprocessing steps to attenuate them. We validate our approach through a group analysis of six awake subjects performing awake tasks in a long-block design, and we compare our results to DES. Our results demonstrate a previously undescribed thermal impulse response function for infrared thermography. We discuss the implications of this response function for IRT mapping and its relationship with analogous phenomena in fMRI.

## 2. Methods and Materials

### Equipment

A novel thermography system was designed for awake functional mapping (Figure 1). The system centerpiece is a FLIR T1020sc thermal camera (resolution 1024 × 768, framerate 30 Hz, NETD < 20 mK). The camera is supported by a tripod with a horizontal extending arm, all of which are enveloped by a sterile plastic cover during data collection. A data transfer wire extends from the camera along the operating room floor to a computer workstation housed in a mobile computer cart. The role of the workstation is data storage, processing, visualization, and integration of the auxiliary devices for task administration and behavioral monitoring. The computer has a speaker for transmitting audio cues to guide the patient through tasks. A microphone attached to the computer cart records all task audio. The computer is connected via Bluetooth to a tablet device which may deliver visual stimuli to the patient, as well as a haptic glove which is worn during hand motor or sensory tasks. This haptic glove tracks joint angles of the hand which may be used to reconstruct the hand position and may deliver vibrotactile stimulation to the fingertips for hand sensory tasks.

We have also developed a custom software application for the thermography system. The application has a live data feed from the thermal camera which aids the camera operator in optimally positioning, orienting, and focusing the camera over the craniotomy during setup. The application connects to and time-synchronizes the thermal camera with all auxiliary devices. This includes starting and stopping thermal camera recording, calibrating thermal detector drift (non-uniformity correction) between task epochs, and issuing all task-related stimuli. The application is equipped with a training mode to teach patients how to perform the task prior to surgery. A free text section allows the operator to save any relevant observations or patient metadata related to the experiment at recording time. Following each epoch, the thermal data and timings are saved locally for further analysis. While the thermography system has been designed for real-time data processing and analysis, this work focuses on the analysis steps needed so that the real-time result is reliable. As a result, the results presented in the sections below are all post hoc analyses.

## 3. Data Collection

Patients undergoing awake craniotomy with DES functional mapping for glioma resection were recruited for intraoperative thermal imaging. Children and pregnant women were excluded from the study. All patients underwent preoperative neuropsychological evaluation to exclude patients with severe pre-existing functional or behavioral deficits. All study protocols were approved by the corresponding institutional review board. Participation in the study did not prolong operation time by more than fifteen minutes. No changes were made to standard surgical workflow, such as size and shape of craniotomy, anesthesia protocol, or number of DES stimulations. Detailed procedures for awake craniotomy with stimulation mapping have been described previously [3]. In brief, all patients underwent an asleep–awake–asleep sedation protocol per routine. Initially, patients were sedated via titration of intravenous propofol and remifentanil. Following standard craniotomy, all sedation was held. The dura was opened sharply and, upon exposure of the cortical surface, direct electrical stimulation (DES) mapping was performed using 60 Hz, 1–4 mA pulses of 1 millisecond duration.

Mapping with infrared thermography (IRT) followed DES mapping. The infrared camera and tripod were wrapped in a polyethylene surgical cover, and wheeled over until adjacent to the surgical bed. Only one cover was used per patient, and it was not moved during recording. The camera was then carefully positioned over the craniotomy and adjusted as needed using the live camera data feed on the computer monitor as a guide. The haptic glove was then placed on the patient if a hand-related task was being performed. Infrared thermography was performed using a long-block design. Each patient was assigned to one of three mapping tasks—hand motor, hand sensory, or face motor—based on expected exposure of functional areas within the craniotomy. The task was explained to each patient prior to mapping, and each patient practiced the task at least once prior to mapping.

Mapping took five minutes for each patient, consisting of ten epochs of thirty seconds each. Patients participated in a task stimulus at the start of each epoch. Hand motor patients performed a hand clench, while face motor patients performed a lip purse. In either case, three fast beeps (one second duration) were played to prompt the patient to hold the position. Two seconds later, two slower beeps were played (one second duration) to release the position. For hand sensory patients, the patient remained at rest while receiving a two-second pulse of vibrotactile stimulation to the fingertips. Patients then remained at rest for the remainder of the epoch. Patients were observed while performing the task to ensure high-quality participation. Patients were not engaged during rest time except if they appeared to be falling asleep.

## 4. Data Analysis

Thermography data were primarily analyzed using principal component analysis (PCA) and group PCA to reveal patterns of stimulus-dependent temperature changes and their spatial distribution across patients. However, several preprocessing steps were applied prior to PCA to account for artifacts in thermal data. Our overall procedure is detailed here, while additional details and justification on specific subroutines are described in their respective sections below. Preprocessing begins with motion correction to bring all frames from each patient into the same spatial reference frame. A craniotomy mask is manually drawn over the craniotomy in the reference frame. Data outside this mask were excluded from all analysis in order to limit the effects of non-brain pixels on the functional analysis. Each masked frame of thermal data was normalized to account for thermal drifts over time, and then the time series of each pixel within the mask was normalized to account for varying signal amplitudes over space. Data from all epochs were averaged together, which underwent frame and pixel normalization again for PCA stability. Finally, the rapid cooling effects of air currents were attenuated before the final PCA analysis.

### 4.1. Motion Correction

The craniotomy and brain surface exhibit rigid and nonrigid motion due to a variety of factors, including the cardiorespiratory cycle and patient movement. As the temperature gradients with respect to space typically exceed the temperature gradients with respect to time, motion-related temperature changes become the dominant signal, interfering with subsequent analysis. Our approach to motion correction has been described previously [25], in which frame-to-frame motion is modeled as a two-dimensional spline function. We expand this approach further by implementing a pyramid approach, where the deformation field is first estimated on a downsampled version of the image and then upsampled as initial conditions for the final calculation. We estimate performance of the motion correction algorithm using image-quality metrics, which are calculated for each frame of thermal data before and after motion correction using the first data frame as a spatial reference. Three image-quality metrics were used: the image mean squared error (MSE), the peak signal-to-noise ratio (PSNR), and the structural similarity index metric (SSIM).

### 4.2. Global Thermal Drift

The overall craniotomy temperature may change slowly over the course of the experiment due to exposure to the external environment. These changes are independent of the thermal response to functional activation and may confound subsequent analysis if the magnitude of the drift is similar to the magnitude of task-induced thermal changes. We account for this through a baseline subtraction and normalization approach. Only pixels within the craniotomy mask are considered for this analysis. The median temperature is calculated for each frame, and this value is subtracted from all pixels in the frame. Next, the median absolute deviation is calculated for each frame, and all pixels in the frame are divided by this value. Medians are chosen over means to mitigate the impact of outlier pixel temperatures on thermal drift correction, such as pixels near the craniotomy edge or any surgical objects on the brain surface.

### 4.3. Amplitude Normalization

Local perfusion is a significant contributor to brain surface temperature. Baseline pixel temperature and the amplitude of temperature fluctuations are therefore modulated by the local vasculature. For example, pixels near large surface vessels are considerably warmer, while pixels near the craniotomy edge are often considerably colder. If the variance of pixels is not normalized prior to principal component analysis, the results will be biased towards thermal patterns from higher-variance pixels. This may over-represent vessels in the final component maps at the cost of activated tissue. We account for this by normalizing the amplitude of the time series of each pixel temperature. We use an identical procedure as the thermal drift correction specified above, but applied to each pixel time series rather than each frame. As a result of normalization, the time series median is set to zero and its median absolute deviation is set to one.

### 4.4. Air Current Denoising

The neurosurgical operating theater is ventilated with cold air for sterility requirements [26,27]. The interaction of air currents with craniotomy motion creates turbulence which drives erratic cooling patterns on the brain surface. This introduces substantial noise into the pixel-level signal. We propose a simple filter for air current attenuation of pixel time series.
$$y=\left(x*e^{x}\right)/\left(w*e^{x}\right)$$

Here, x is a time series of thermal data from one pixel, $e^{x}$ is the element-wise exponent of the time series x, w is a discrete Gaussian window function with a standard deviation of σ, and y is the reconstructed signal. Operators represent convolution and element-wise division. Outputs of both convolutions were cropped to match the size of the input time series x. The filter may be understood as a sliding window softmax filter with an additional Gaussian scaling coefficient. In practice, it functions as a moving average filter which selectively preserves peaks and attenuates troughs. This is effective for air currents which induce rapid momentary cooling by interpolating data from nearby local maxima.

### 4.5. Principal Component Analysis

Principal component analysis (PCA) is a dimensionality reduction technique where the input data are transformed such that the dataset variance is primarily explained along the orthogonal principal component vectors. As applied to our analysis, each pixel time series is treated as an independent data point, so principal components represent patterns of temperature change over time which are common across pixels. Loading coefficients represent the contribution of each principal component to each pixel time series. A component map can then be created for each component time series where the pixel value is the loading coefficient. For group PCA analyses involving multiple subjects, the data from all subjects were spatially concatenated before PCA was applied. Next, the weighting of each subject’s data was adjusted to be proportional to the inverse of the number of pixels contained in each subject mask. Following group PCA, the data were split to achieve individual spatial components which share a time series.

## 5. Results

Six patients were recruited for intraoperative functional mapping studies with IRT, consisting of two patients for each of three tasks: hand motor, hand sensory, and face motor. Patient demographics and task assignment is shown in Table 1 below. Four of the patients were female (67%) and two of the patients were male (33%). The average patient was 37.3 years old with a standard deviation of 7.2 years.

Data from all patients were preprocessed as specified above. Data were downsampled to 5 Hz and cropped to an epoch duration of 20 s. All data were used in the analysis except the first epoch from patient 4, which was excluded due to a sudden large motion event. Optimal motion correction parameters have been found previously for this downsampling rate: image downsampling rate of 4, grid downsampling rate of 4, regularization coefficient of 0.008, and four optimization steps per frame [25]. These parameters were used for the base pyramid level, for which the upper pyramid level doubled both downsampling rates. Image quality following motion correction was measured and the aggregate results are displayed below (Figure 2). Motion correction improved median image quality across the entire epoch for all three image-quality metrics.

Following preprocessing, all epochs were averaged together and analyzed with PCA. Two experiments were performed: one in which PCA was applied to one subject at a time (Individual PCA), and one where PCA was applied to all subjects simultaneously (Group PCA). The results of these analyses are described below. Complete functional keys and original full-color DES photos are available for inspection in the Appendix A, along with all component maps for each patient.

Component time series from group PCA are shown in Figure 3. Five components explained a total of 87% of the variance, with 34% contained in component 1, 19% contained in component 2, 13% contained in component 3, 12% contained in component 4, and 9% contained in component 5. The time series of the first two components were notable. Versions of both of these time series were observed in each patient’s individual PCA component time series. Component 1 decreases initially until five seconds, slowly increases until ten seconds, then rapidly increases to its peak at fifteen seconds. Component 2 is stable for the first three seconds, then increases to its peak at ten seconds, after which it stabilizes around seventeen seconds. Specifically, component 2 peaks as component 1 begins its rapid increase, and component 1 peaks as component 2 is nearly stabilized. The remaining components were sequentially higher in frequency and lack clear interpretation at this time.

Loading maps for component 1 were well correlated with functional areas for two patients, partially correlated for two patients, and anticorrelated in two patients (Figure 4). For patient 1 (lip pursing), F and G were positive for speech arrest, but only G is covered by the map. For patient 2 (lip pursing), H and I (hesitation) are covered but J (dysarthria) is not. In addition, there is vast activation over the unlabeled cortex. For patient 3 (hand clench), there is overlap in terms of hand-motor labels (D, E). For patient 4 (hand clench), the entire hand motor and sensory cortex is unactivated, with strong activation nearly everywhere else. For patient 5, (finger sensory), label 5 (finger tingling) is unactivated. For patient 6 (finger sensory), there is strong coverage over the finger numbness areas (D–E). When considered as a functional indicator across patients, component 1 maps do not consistently overlap with task-specific functional areas.

Loading maps for component 2 were well correlated with functional areas overall (Figure 5). In patient 1 (lip pursing), both F and G (speech arrest) locations were active, along with a large patch over label B (lip sensory), which is an unsurprising secondary activation. In patient 2 (lip pursing), the largest positive area lies over J (dysarthria). There are some moderate activations overlapping areas which induce hesitation when stimulated (H, G). In patient 3 (hand motor), the hand-specific labels (D, E) are both covered by component 2. While label coverage here appears to be edge artifact, this is a result of thresholding for this figure and the loading magnitudes are especially high for these areas. In patient 4 (hand motor), there is strong activation across the hand motor and sensory areas (all labels). In patient 5 (finger sensory), there is activation over label 5 (finger tingling) as well as the rest of the sensory cortex (1–8), while the motor cortex (9–13) is unactivated. In patient 6 (finger sensory), D and E (finger numbness) were strongly activated.

Component 2 maps also exhibited activation in unlabeled areas. For example, in patient 3, there is a large area of positive cortex adjacent to and encompassing label 4 (mouth motor) that may indicate activation in the premotor cortex. This spatial activation pattern is also observed in patient 2 and to a lesser extent in patient 4. Our DES testing protocol did not include interruption tasks and would therefore not be sensitive to secondary functional areas, so there are no labels here. While this may explain some of the activations in the motor tasks, it does not explain the additional activations in the passive finger sensory tasks (patients 5 and 6). In addition, all maps suffer from some artifact along the edges and small sparse activations throughout.

## 6. Discussion

We have uncovered in component 2 a pattern of temperature change which localizes to DES-confirmed functional areas across patients and tasks. This component was observed at the group level but is also found at the individual level across patients. The time course of this component is significant for its structural similarity to the BOLD hemodynamic response function (HRF) found in functional MRI. The time course of component 2 has an initial dip, broad peak, and a small post-stimulus undershoot akin to the HRF. While the thermal time course is notably slower than the BOLD HRF, this is expected as increased blood flow is a prerequisite for tissue heating, which is not an immediate process [19]. Similarly, it takes some time for the tissue to cool post-activation. We define this time series as the thermodynamic response function (TRF) due to its similarity to the BOLD HRF, functional localization, and likely role as an impulse response for IRT.

We have also found a primary component which is stronger in magnitude than the TRF and is also present in every patient. While it lacks any consistent spatial specificity with DES results, the time course suggests that it may represent hemodynamic patterns following the TRF. Local perfusion increases in functional areas via arterial dilation, which constricts once the stimulus event is over. The larger spatial coverage primary component may represent the spread of the venous network as it drains the increased amount of blood to the region, which causes heating in regions not directly associated with the activation. This is known as the draining vein problem in fMRI [28,29]. While in fMRI the problem goes away as the activated bolus becomes diluted downstream (within 5 mm), the issue in thermal imaging is worse as the heated blood conducts this heat to adjacent tissue and stores it there. The time constant of this transfer and its decay are not known but the data seem to suggest that it is much slower to dissipate, leading to the larger activation fields in the primary component.

We are the first to demonstrate a coherent TRF impulse response across subjects. Prior studies relied on temperature thresholds in order to distinguish functional from nonfunctional tissues during classification. This is less reproducible and less robust than impulse-based mapping, as absolute temperature values are dependent on many factors including air temperature and humidity, operating room ventilation system type and positioning, type and dose of anesthetic [24,30], neurovascular coupling, and craniotomy size and orientation. Due to the high sensitivity of IRT to functional areas, it may be useful as a screening tool in conjunction with DES confirmation as a way to reduce the stimulation search space and ultimately lower the mapping time. Although the task-related specificity was low in some patients, much of the extra activation was also in the eloquent cortex. Stimulating these areas is also informative for intraoperative mapping.

While there have been many modalities developed for intraoperative functional mapping [31,32], DES has remained at the forefront. Somatosensory evoked potentials are effective for mapping the central sulcus or thalamocortical tract [33,34] but has limited general applicability and its accuracy may vary based on lesion location [35]. Additionally, electrocorticography has been used for some time to track discharges during epilepsy surgery [36] and is also viable as an intraoperative mapping tool for complex functions [37,38]. Electrode-based methods (including DES) are generally successful due to their basis in neural electrophysiology; however, they share weaknesses in low spatial resolution and low cortical coverage. Optical methods such as IRT compensate for these weaknesses by mapping the entire craniotomy simultaneously with high spatiotemporal resolution. Further refinement of efficient optical methods for brain mapping may develop a synergistic relationship between the two classes of techniques.

## 7. Limitations

We encountered a couple of limitations while performing this study. Foremost, we encountered difficulty in recruiting a large cohort of subjects. Awake craniotomies for glioma resection are relatively infrequent, and not all of these patients are strong candidates for research. Since no alterations to the clinical procedures could be made as a result of participation in the present study, neuroanesthesia protocols could not be adjusted and some patients were unable to stay awake during testing. This resulted in inconsistent or partial task participation which is not suitable for inclusion in the above analyses. Second, measured surface temperature is a result of many factors, of which local functional activation is a major determinant. However, some patients may have local alterations in neurovasculature due to natural variation or glioma invasion. This may alter sensitivity and specificity of thermal mapping in some subjects. Lastly, the vascular network is interconnected so heating or cooling may happen in unexpected areas due to upstream or downstream effects. This may be accounted for by studying the order in which areas experience similar temperature patterns, but our analysis here does not have the capacity to study delays.

## 8. Conclusions

This study analyzes the effects of stimuli on intraoperative brain surface temperature. We have demonstrated a new technique for functional brain mapping using patterns of relative temperature change as opposed to absolute temperature changes. We have developed new hardware and software approaches to support this analysis in the operating room. Using this technology, we were able to observe an impulse response for thermography-based functional mapping. This function is likely the direct thermal byproduct of neurovascular coupling and has strong agreement with positive areas from direct electrical stimulation mapping. These contributions together modernize the current approaches for functional mapping with infrared thermography. We have taken steps towards standardizing the mapping methodology, which will ultimately be necessary for adoption of thermography for real-time surgical use.

## 9. Patent

United States patent pending.

## Figures and Tables

**Figure 1 brainsci-13-01091-f001:**
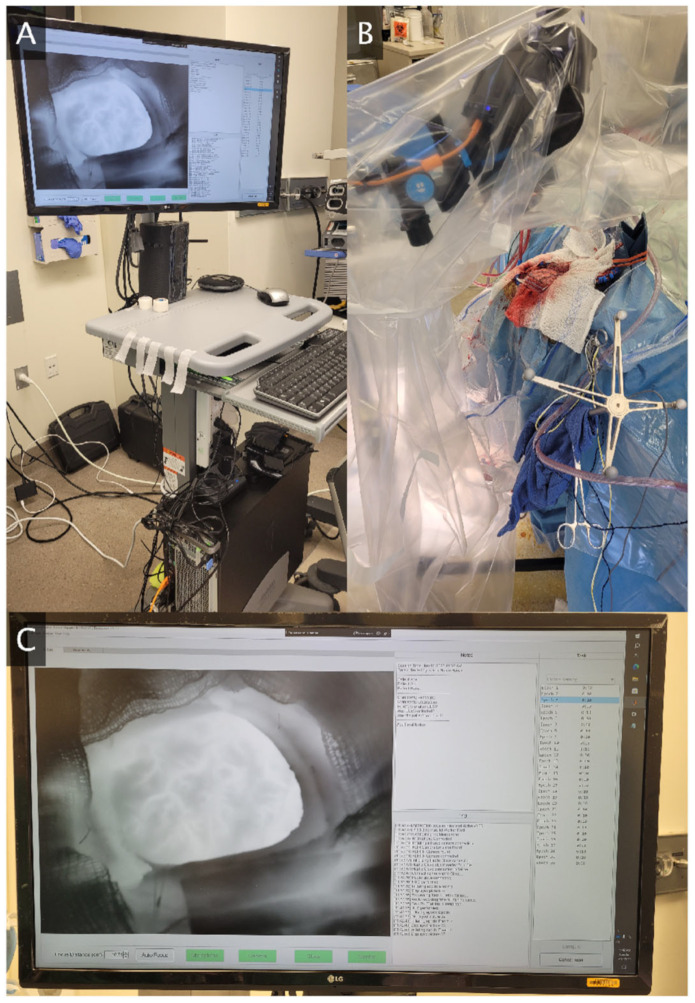
Intraoperative Thermography System. (**A**) Mobile computer cart with workstation computer during intraoperative data collection. The imaging app is visible on the screen with a real-time feed of craniotomy data. (**B**) Infrared camera positioned over the craniotomy during data collection. The camera and support system are wrapped in a sterile plastic cover. (**C**) Close-up of thermography task administration and data collection software.

**Figure 2 brainsci-13-01091-f002:**
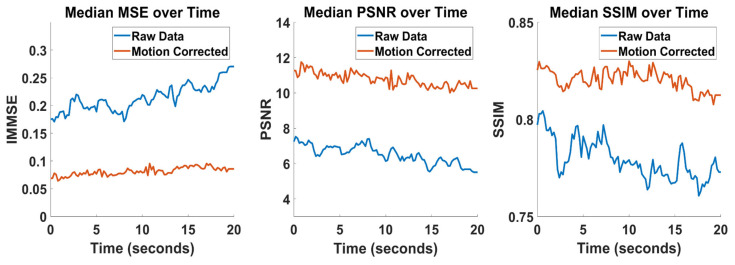
Motion Correction Quality. Image quality before and after motion correction was estimated for all subjects. Aggregate values were computed by taking the median across all subjects and all epochs. The first frame of the subject’s data was used as the reference frame for all other epochs (full coregistration). Image quality is plotted over time for the mean squared error (MSE), peak signal-to-noise ratio (PSNR), and structural similarity index metric (SSIM).

**Figure 3 brainsci-13-01091-f003:**
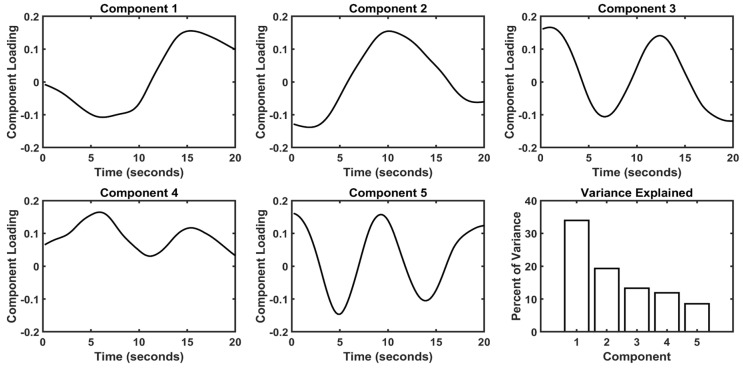
Group PCA Time Series. The loading time series and percentage of variance explained are shown for each component. Components are displayed in order of decreasing variance. Stimulus onset was between zero and one seconds for all tasks, and stimulus offset.

**Figure 4 brainsci-13-01091-f004:**
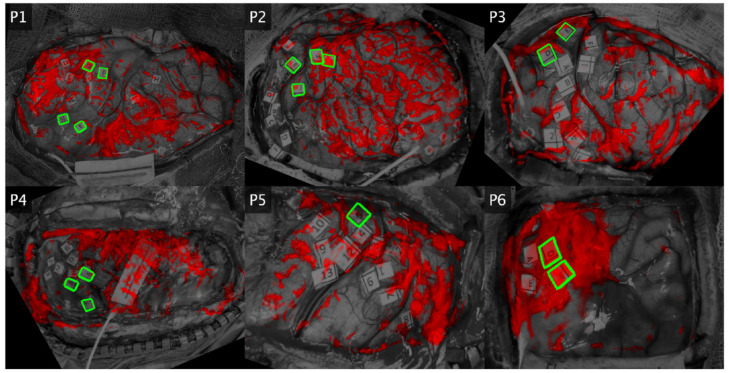
Component 1 Group PCA Maps by Patient. Spatial loadings are shown for the second component in each patient. The craniotomy and DES labels are shown as a grayscale image. Red overlay indicates where the map component exceeded the mean value and is bright red where the loading exceeds two standard deviations above the mean. Black areas indicate parts of the infrared image outside of the field of view of the white-light image. Green indicates the edges of positive DES labels which are directly related to the IRT task.

**Figure 5 brainsci-13-01091-f005:**
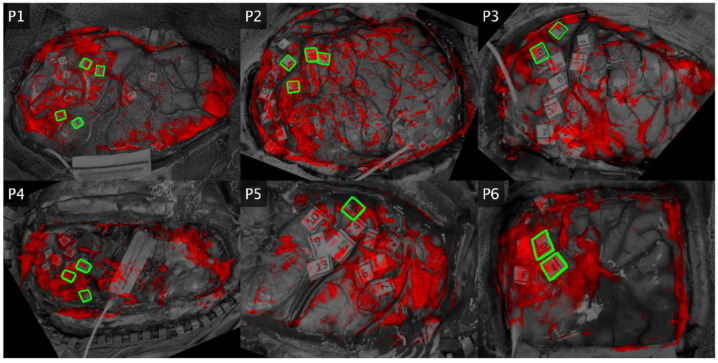
Component 2 Group PCA Maps by Patient. Spatial loadings are shown for the second component in each patient. The craniotomy and DES labels are shown as a grayscale image. Red overlay indicates where the map component exceeded the mean value and is bright red where the loading exceeds two standard deviations above the mean. Black areas indicate parts of the infrared image outside of the field of view of the white-light image. Green indicates the edges of positive DES labels which are directly related to the IRT task.

**Table 1 brainsci-13-01091-t001:** Patient Information. Age, sex, and task performed for each patient.

Patient	Task	Age	Sex
1	Lip Purse	46	Male
2	Lip Purse	41	Female
3	Hand Clench	31	Male
4	Hand Clench	44	Female
5	Finger Sensory	33	Female
6	Finger Sensory	29	Female

## Data Availability

The data presented in this study are available on request from the corresponding author. The data are not publicly available due to patient privacy concerns.

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
