# Peer review of "A Novel Intraoperative Mapping Device Detects the Thermodynamic Response Function"

_brainsci, 2023, doi:10.3390/brainsci13071091_

Round 1

Reviewer 1 Report

Firstly, the study is about a very important field in both. research and clinical practice. I have a few questions to address.

The Operation Room is a massive, dynamic (even chaotic) theater.  It would be very wise to provide the background temperature and humidity recordings. Was there a heater below the patients? What were the patients’ core body temperature, and peripheral body temperature?

It would be very good to provide a similar figure (fig 2, method performance over time) of the group’s former study (ref 25).

The sterile plastic covers may change from one trial to another, the details are better provided.

The limitation of the study also comes from the design as all patients had at least three different sets of tasks to perform. This vastly increases the complexity of the data and the interpretation of the outcomes. Furthermore, the PCA 2 and the other components are very coarse sine waves where the smearing may have obscured the results.

Finally, the natural vascular variations and the microvessels could have shadowed some of the outcomes. This should also be noted as a limitation.

The number of patients seems to create a kind of ambiguity of the results. Perhaps it is best to add phrases like “preliminary results” or “early phase experiment outcomes” etc.

Author Response

Firstly, the study is about a very important field in both. research and clinical practice. I have a few questions to address.

The Operation Room is a massive, dynamic (even chaotic) theater.  It would be very wise to provide the background temperature and humidity recordings. Was there a heater below the patients? What were the patients’ core body temperature, and peripheral body temperature?

We have not recorded the background temperature and humidity, nor the patient’s core or peripheral body temperatures. There was no heater below the patients. Our focus is more on temperature changes, rather than the absolute baseline temperature. Baseline temperature can fluctuate in patients for many factors, many of which are out of our control. For example, the temperature and humidity are controlled for sterility requirements. We note that the thermoregulation of the brain is largely independent of thermoregulation of the rest of the body. There is a nice review from 2019 in Neuron about this (PMID: 29621489). 

It would be very good to provide a similar figure (fig 2, method performance over time) of the group’s former study (ref 25).

This is a great suggestion. As the method has been modified slightly since our previous work it is reasonable and justified to confirm it is still working well. We now provide motion correction validation figures in the results.

The sterile plastic covers may change from one trial to another, the details are better provided.

Only one sterile cover was used per patient. Once it was placed at the start of the experiment it was not moved until the experiment was complete. Different covers were used in different patients for sterility requirements. We have added a sentence to the data collection section to clarify this. 

The limitation of the study also comes from the design as all patients had at least three different sets of tasks to perform. This vastly increases the complexity of the data and the interpretation of the outcomes. Furthermore, the PCA 2 and the other components are very coarse sine waves where the smearing may have obscured the results.

Including data from three different sets of tasks makes the analysis more challenging to perform. Despite this, we were able to demonstrate regional functional sensitivity in component two across patients. This suggests that the functional component presented here is invariant to the task being performed. We see this as a strength, not a limitation. 

Our air current attenuation procedure introduces a smoothing effect to account for noise from air currents, however the design of the filter limits the smoothing interval to about one second. Group PCA also has an innate smoothing effect as it looks for similar changes in variance across subjects, thereby smoothing out patient-level variance. Individual PCA results (available in the supplemental information) do not in general have the same very coarse sine wave appearance, suggesting that the group PCA is the primary reason the components have the smooth appearance. If a signal property is the result of aggregating subjects, it likely represents a genuine underlying signal as opposed to smearing which obscures results. We do not doubt that there are other higher-frequency signals occurring at smaller scales, but the functional relevance and consistency of these signals across patients have yet to be demonstrated. 

Finally, the natural vascular variations and the microvessels could have shadowed some of the outcomes. This should also be noted as a limitation.

This is a great point. We believe that there is likely some changes in thermal signals as a result of the positioning of functional regions with respect to local vasculature. Due to subject-level variance in vascular structure and plasticity, the same task may have differing results. We now mention this in the new limitations section. Our amplitude normalization technique partially addresses this issue and the corresponding section has been 

The number of patients seems to create a kind of ambiguity of the results. Perhaps it is best to add phrases like “preliminary results” or “early phase experiment outcomes” etc.

Recruiting awake craniotomy patients is challenging, and many of the papers in the literature surrounding infrared thermography in neurosurgery have fewer patients than we do. However, we share your concern and agree that further study of the presented phenomena with more patients would be helpful. We now discuss these issues in the new limitations section. 

Reviewer 2 Report

In this study, authors reported a novel intraoperative mapping device detects the thermo dynamic response function”. The overall contents of this manuscript is not well organized to give a clear overview of this work. I have suggested some major comments about this work are as the following:

Comments to the Authors:

1.     Authors should revised the abstract of this study with clearly explain the objective of this study, method, results and conclusion of this study. The current form of the abstract is very weak.  Authors should write the abstract clearly including background with previous study unsolved questions and current study address/solve the problem, method, results with numerical values and significance difference.

2.     Authors presented six patient datasets; it should be explain how many male and how many female with ages.

3.     Authors should revise the introduction section in three paragraph, first paragraph for medical intraoperative mapping devices; second paragraph for past research based on intraoperative mapping device detects the thermo dynamic response in patients, third paragraph for gap between past and present results and in last paragraph objective of this study and hypothesis.

4.     Author should write the method section in more details with complete description of the statistical analysis.

5.     In Figure 1 author should write the name of each sub-Figure like A, B, C and explain in captions instead of left, right and below.

6.      My suggestion is that the authors should write discussion section clearly in more details like how and why this study is important than previous studies based on intraoperative mapping device detects the thermo dynamic response in patients.

7.     The authors should write some limitations of this modal in details and future applications in more details.

8.      Authors should write the conclusion of this study.

 Moderate editing of English language required. 

Author Response

In this study, authors reported a novel intraoperative mapping device detects the thermo dynamic response function”. The overall contents of this manuscript is not well organized to give a clear overview of this work. I have suggested some major comments about this work are as the following:

Comments to the Authors:

  1. Authors should revised the abstract of this study with clearly explain the objective of this study, method, results and conclusion of this study. The current form of the abstract is very weak.  Authors should write the abstract clearly including background with previous study unsolved questions and current study address/solve the problem, method, results with numerical values and significance difference.

The abstract has been modified to be stronger. We are more direct about the objective of the study and how our work addresses gaps in the field.

  1. Authors presented six patient datasets; it should be explain how many male and how many female with ages.

We have added a table with additional information on the patients. 

  1.     Authors should revise the introduction section in three paragraph, first paragraph for medical intraoperative mapping devices; second paragraph for past research based on intraoperative mapping device detects the thermo dynamic response in patients, third paragraph for gap between past and present results and in last paragraph objective of this study and hypothesis.

Our introduction is written from the neurosurgical perspective, as our paper concerns a novel device which we hope will one day be used by neurosurgeons. The current organization of the introduction already closely resembles the reviewer’s suggested layout, with some differences that are difficult to implement due to the limited availability of past studies in this area. We address each point more specifically below:

  1. The reviewer requests the first paragraph be on medical intraoperative mapping devices. Currently direct electrical stimulation (DES) is the gold standard and is by far the most commonly used technique. There is a paucity of available technologies in this area, so we focus on DES. This paragraph exists in the original manuscript, but it is placed second, preceded by a background paragraph on glioma surgery which we believe is helpful for most readers in understanding the scope and impact of our work.
  2. The reviewer requests the second paragraph be on “past research based on intraoperative mapping device detects the thermo dynamic response in patients”. In the present manuscript, we have included this as the third paragraph, which surveys the available literature on functional mapping using brain surface temperature. 
  3. The reviewer requests that the third paragraph be on the gap between past and present results. This is vague, but we believe this is encapsulated in our fourth paragraph, which elucidates the gap between previous IRT results and what is needed for neurosurgery, which is the set up for what our device attempts to accomplish.
  4. The reviewer suggests that the last paragraph describes the objective of this study and the hypothesis. We are clear about the study objective; the last paragraph begins with “The primary goal of this work…”. Our study is constructed as data-driven as opposed to hypothesis-driven, which we believe is appropriate for fair assessment of a novel device.

In summary, the introduction layout is similar to that requested by the reviewer, with the exception of a preceding first paragraph with background on glioma surgery. We decline to remove this paragraph as it is fundamental to the purpose of our research.

  1. Author should write the method section in more details with complete description of the statistical analysis.

We have added additional details to the methods section.

  1. In Figure 1 author should write the name of each sub-Figure like A, B, C and explain in captions instead of left, right and below.

Figure 1 has been reorganized with letter labels.

  1. My suggestion is that the authors should write discussion section clearly in more details like how and why this study is important than previous studies based on intraoperative mapping device detects the thermo dynamic response in patients.

Please take note of the first sentences of the third paragraph: “We are the first to demonstrate a coherent TRF impulse response across subjects. Prior studies relied on temperature thresholds in order to classify functional from nonfunctional tissues”. This field is unfortunately small so there are not many papers to compare to. We expand on the paper impact further in the conclusion section.

  1. The authors should write some limitations of this modal in details and future applications in more details.

We now include a specific section on study limitations..

  1. Authors should write the conclusion of this study.

We have now included a conclusion section.

Reviewer 3 Report

Although, the infrared mapping looked good, Figure 3 was not impressive since the functional areas did not correlate well with IRT task as described in the text since only two of the subjects showed a good correlation. 

2)  The authors should show the correlation of their IR technique with BOLD functional MRI and present evidence.

Author Response

Although, the infrared mapping looked good, Figure 3 was not impressive since the functional areas did not correlate well with IRT task as described in the text since only two of the subjects showed a good correlation. 

The results in Figure 3 show the spatial map of component 1 from the group PCA analysis. We agree, and describe in the text, how the spatial correlation of this component with the DES areas is poor. That is not what is interesting about this component. It is interesting that this component is present strongly in all patients in individual PCA, and is present with higher variance explained than the proposed functional component in the group analysis (see Supplemental Information). Neither components have been described quantitatively in the literature on functional imaging with infrared thermography. 

We include it for a few reasons. First, its robustness and time course point to interesting physiological sequelae to brain activation. Second, it elucidates why simply looking for brain temperature increases, as is often done in previous studies, is likely an inferior technique to analysis of temperature patterns (e.g. PCA), since this component would confound the first analysis and not the second. Third, the spatial patterning of this component versus the initial one suggests it may be related to the draining vein problem in fMRI. The spreading of brain temperature has a wave-like characteristic similar to the observations made by Shevelev and Tsicalov on fast thermal waves (PMID: 9015336).

2)  The authors should show the correlation of their IR technique with BOLD functional MRI and present evidence.

At present, DES is the gold standard for functional mapping in the operating room, and so we use it as our primary comparison metric. However, we do think it is interesting to consider the relationship to BOLD fMRI and plan to work on this in the future. There are a couple of issues that need to be solved before the comparison can be made:

  1. The fMRI tasks need to be slowed down to match the pacing of the IR tasks (20 second stimulation epochs). This is not currently done during preoperative functional mapping and we would need separate research scans for this. 
  2. Spatial registration of the IR images and the fMRI space needs more work. A previous paper exists but shows mean accuracy of about 3 mm, which will muddy the analysis considerably. We are working on solving this presently.

Due to the methodologic complexity of the analysis, as well as needing to acquire a separate dataset, we believe the fMRI comparison would be better presented in a separate paper as including it distracts from the primary message of this one. 

Round 2

Reviewer 1 Report

The authors mostly responded to former review points.

Reviewer 3 Report

The authors have responded to my issues and I feel that article is now ready for publication.  They have deferred some analyses that I suggested for a future paper which is understandable.